# A Novel Filtering Method of 3D Reconstruction Point Cloud from Tomographic SAR

**Shuhang Dong** [1,2]**, Zekun Jiao** [2]**, Liangjiang Zhou** [2,3,]*****, Qiancheng Yan** [1,2] **and Qianning Yuan** [1]

1   School of Electronic, Electrical, and Communication Engineering, University of Chinese Academy of Sciences, Beijing 100049, China
2   National Key Laboratory of Microwave Imaging Technology, Aerospace Information Research Institute, Chinese Academy of Sciences, Beijing 100190, China
3   Qilu Research Institute, Institute of Aerospace Information Research, Chinese Academy of Sciences, Jinan 250101, China
*   Correspondence: ljzhou@mail.ie.ac.cn

**Abstract:** With the development of airborne synthetic aperture radar (SAR) technology, the 3D SAR point cloud reconstruction has emerged as a crucial development trend in the current SAR community. However, due to measurement errors, environmental interference, radar decoherence, and other noises associated with the SAR system, the reconstructed tomogram is often deteriorated by numerous noisy scatterers. As a result, it becomes challenging to obtain high-quality 3D point clouds of the observed object, making it difficult to further process the point cloud and realize target identification. To address these issues, we propose a K nearest neighbor comprehensive weighted filtering algorithm. The filtered point cloud is evaluated quantitatively using three-dimensional entropy. In this study, we adopted various filtering methods for simulated data, P-band data of Genhe, and Ku-band data of Yuncheng to refine the tomogram and compare their performances. Both qualitative and quantitative analyses demonstrate the superiority of the filtering algorithm proposed in this paper.

**Keywords:** airborne tomographic SAR; compressive sensing 3D reconstruction; point cloud filtering; three-dimensional entropy

## 1. Introduction

Synthetic Aperture Radar (SAR) [1,2] is an active remote sensing technique which aims to extract the scattering information of the observed object by transmitting electromagnetic waves. In contrast to optical imaging, SAR works all weather and all day. Consequently, SAR can achieve over-the-horizon and high-resolution earth observation, making it a vital tool for remote sensing applications. Currently, SAR has become an important technical means of remote sensing and shows great potential for a wide range of applications, including target detection, 3D reconstruction, disaster assessment, etc.

Due to the differences between the mechanism of SAR and optical imaging, compared with the simplicity of optical images, 3D objects will introduce aliasing on 2D SAR images in areas with steep terrain and complex environments. As a result, SAR encounters numerous challenges in target recognition and image interpretation, which has become a bottleneck restricting its performance and further development. SAR three-dimensional imaging can reconstruct the three-dimensional electromagnetic scattering distribution of the target, eliminate the shrinkage, overlapping, top and bottom inversion, and other phenomena caused by the imaging mechanism in SAR images, which is of great significance for target interpretation, urban mapping, and other applications.

Since the last century, interferometric SAR (InSAR) technology has been developed to obtain the three-dimensional information of a target. However, InSAR only uses phase differences observed from different angles or distances; it resolves elevation information

based on these phase differences. In cases where multiple scatterers overlap in the same pixel, InSAR can only determine the position of the composite scattering center and cannot realize three-dimensional resolution ability. With the multi-baseline SAR system, radar tomography makes it possible to distinguish multiple overlapping scatterers in a single pixel by obtaining multiple SAR images of the same scene and reconstructing the elevation profile. This method achieves true 3D SAR imaging [3–6].

The use of 3D point cloud reconstruction technology has gained attention in many applications, such as developing 3D models of urban buildings, equipment structures, and historical relics [7–9]. These point clouds can be extracted from various sources, including 3D laser scanners, laser radars, oblique photography, and the tomographic SAR-(TomoSAR) 3D reconstruction discussed in this paper. Consequently, processing and interpreting point cloud data has become a focus in various applications. However, due to the influence of equipment limitations, external environment, target structure, and decoherence, the reconstructed point cloud inevitably contains noise and distortions that directly affect subsequent processing tasks, such as data registration, feature extraction, surface reconstruction, and 3D visualization. Therefore, effective point cloud filtering is crucial. The reliability of filtering algorithms for point cloud data has become a hot topic of research aiming to ensure appropriate subsequent processing.

In the process of two-dimensional SAR imaging, motion compensation error, approximation error of the imaging algorithm, and other factors can lead to poor coherence between the acquired two-dimensional images. During three-dimensional reconstruction by tomographic SAR, part of the noisy scatterers will emerge in the upper and lower positions of the target. In addition, random noise will also form many discrete scatterers in the point cloud. These noises are difficult to estimate and compensate, so it is necessary to improve the 3D reconstruction effect through the post-processing of point clouds.

Current filtering algorithms for 3D point clouds can be roughly divided into three categories [10,11]: algorithm based on the attribute characteristics of point clouds, algorithm based on the spatial structure of point clouds, and other point cloud filtering methods.

The first kind of filter is to set the range or to set conditions based on the attributes of the point cloud, retain the scatterers within the range, or meet the conditions. This type of filter includes a pass-through filter and a conditional-removal filter. Filtering for a 3D reconstructed point cloud based on the amplitude threshold is a typical pass-through filter.

The second type is mainly based on the spatial structure of point clouds, that is, using the distance and spatial relationship between points to achieve point cloud filtering. The representative filters are statistical filters and radius filters with similar principles, which have better filtering effects for outliers. The statistical filter computes the average distance from each point to its K nearest neighbors. Then the distances of all points in the point cloud should form a Gaussian distribution in which points outside of the variance can be removed based on the pre-set threshold.

In addition, there are voxel-grid filters and uniform-sampling filters. Uniform sampling filters generate a sphere with a certain radius and outputs the result as the point closest to the center within each sphere. Their main function is for down-sampling point clouds, while also have the effect of filtering noise points to a certain extent.

The third type of filter is relatively complex, such as project-inliers filtering, model filtering, Gaussian-Kernal filtering, etc. Some of these algorithms require other known model information or are not suitable for complex scenarios. Due to these reasons, they are not suitable for TomoSAR point cloud filtering and no further discussion will be made.

Currently, there are many methods for filtering from different perspectives. In our study, we analyzed the characteristics of various algorithms, compared their performance, and drew on various advantages to propose a filtering algorithm suitable for point clouds by TomoSAR.

## 2. Methods

This section outlines the methodology presented in this paper. Firstly, the principles of 3D reconstruction of tomographic SAR, classical point cloud filtering methods, and the evaluation approach based on 3D entropy are introduced. Next, the entire processing framework is presented, with each step of the method explained explicitly.

### 2.1. TomoSAR 3D Reconstruction

The three-dimensional reconstruction process of TomoSAR can be summarized in the following way. Initially, the original radar echo signals of different incidence angles are obtained. Next, the BP imaging algorithm [12,13] is adopted to realize two-dimensional imaging corresponding to different equivalent antenna phase centers. In addition, an image registration algorithm is carried out to fulfill sub-pixel level image registration. According to the signal model of TomoSAR, a compressed sensing algorithm is used to extract the target scattering distribution in the elevation direction.

Affected by many factors such as orbital wind and atmospheric turbulence, it is difficult to control the flight path accurately. Therefore, it is very hard to obtain high-precision two-dimensional SAR images and achieve sub-pixel registration of SAR images. Compared with other frequency-domain imaging algorithms, the BP imaging algorithm has many advantages, such as good phase retention ability and accurate compensation of the flat phase in the two-dimensional imaging process, which is conducive to TomoSAR preprocessing and to following a three-dimensional reconstruction process.

Just like traditional 2D SAR imaging, the range resolution is achieved by transmitting large bandwidth signals combined with pulse compression technology. Azimuth high resolution is achieved through a synthetic aperture. In order to obtain the elevation resolution, the equivalent array of elevation is constructed by using multiple track observations which meet Nyquist's law to achieve the synthetic aperture and to obtain the resolution ability in the elevation direction. The TomoSAR 3D imaging geometric model is shown in Figure 1 [14,15].

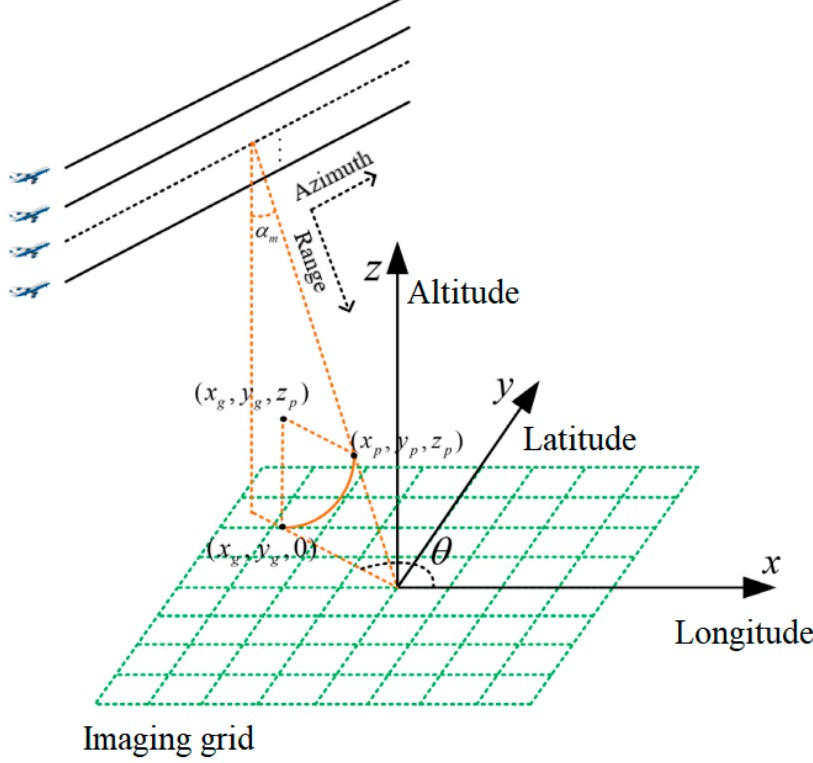

**Figure 1.** TomoSAR 3D imaging geometric model.

As shown in the figure, the observation geometry of TomoSAR is composed of M parallel tracks. When the reference imaging plane is set to the zero-height plane, the BP imaging algorithm is used to obtain the 2D focused images. Based on the 3D imaging geometry, the backscatter coefficient of the observed target is uniformly sampled along with the elevation direction. At this point, the sampling result can be abstracted as the following discrete signal model considering the radar observation noise [16,17]:

$$\mathbf{g} = \mathbf{\Phi} \cdot \mathbf{\sigma} + \mathbf{\varepsilon} \tag{1}$$

In the above equation, $\mathbf{g}$ represents the observation vector of the $M \times 1$ dimensional corresponding to a single range-azimuth pixel of the registered SAR image stack, $\mathbf{\sigma}$ represents the target backscattering coefficient vector to be estimated for the dimension $N \times 1$ in the elevation direction. $\mathbf{\Phi}$ represents the TomoSAR observation matrix of $M \times N$. $\mathbf{\varepsilon}$ is the $M \times 1$ noise vector. $N$ denotes the number of discrete grids in the elevation direction while $M$ stands for the number of repeated tracks.

$$\begin{cases} \mathbf{g} = [g_1, g_2, \cdots, g_M]^{\mathrm{T}} \\ \mathbf{\Phi} = \begin{bmatrix} \exp(-j2\pi\xi_1 z_1) & \dots & \exp(-j2\pi\xi_1 z_N) \\ \vdots & \ddots & \vdots \\ \exp(-j2\pi\xi_M z_1) & \cdots & \exp(-j2\pi\xi_M z_N) \end{bmatrix} \\ \mathbf{\sigma} = [\sigma(z_1), \sigma(z_2), \cdots, \sigma(z_N)]^{\mathrm{T}} \end{cases} \tag{2}$$

TomoSAR 3D imaging technology has shown remarkable performance in complex areas such as cities, but it faces a major difficulty in practical application. Limited by Nyquist's sampling law, using traditional signal processing methods (such as matched filtering, spectrum estimation, etc.) requires tens or even hundreds of repeated flights, resulting in a long data acquisition time and high cost, which not only brings difficulties to data storage and signal processing, but also goes against applications requiring high timeliness.

As the 3D imaging of the tomographic SAR is processed pixel-by-pixel and the ground objects are sparsely distributed in the elevation direction, the compressed sensing theory can be used to reduce the requirements for the number of flights. The Compressed Sensing (CS) theory proposed by Donoho et al., in 2006 [18], on the premise of meeting the sparsity assumption, can realize signal reconstruction with a sampling rate much lower than that required by Nyquist's Law. It is possible to recover the original signal through an appropriate reconstruction algorithm; that is, a small number of radar observations can be used for TomoSAR reconstruction with a high probability. Baraniuk et al. introduced the application of the compressed sensing theory to radar imaging in 2007 [19,20]. Therefore, this article employs the compressive sensing algorithm to accomplish the task of three-dimensional point cloud reconstruction.

### 2.2. Filtering Methods

The reconstructed original 3D point cloud cannot be directly processed and interpreted, because it is inevitably mixed with a large number of noisy scatterers, which must be filtered out by a point cloud filtering algorithm. The causes of these noises are complicated and are affected by various factors such as the equipment itself, the external environment, and the target structure. In the process, baseline error, imaging error, and motion compensation error will result in poor coherence of the acquired 2D image, and thus 3D reconstructed performance will be deteriorated severely. In addition, thermal noise and radar blurring will also produce a certain amount of noise.

In this paper, three traditional filtering algorithms are used to improve the reconstructed point cloud, i.e., amplitude filtering, confidence filtering, and K nearest neighbor filtering. Combining the advantages of the three filtering algorithms, we propose K nearest

neighbor comprehensive weighted filtering. The following is a detailed analysis of these algorithms.

### 2.2.1. Amplitude Filtering

Amplitude filtering is a kind of straight-through filtering, that is, set the filtering threshold based on the attributes of the point cloud, and select the scatterers whose amplitude are larger or less than the preset threshold. The point cloud is filtered according to the corresponding backscattering coefficient of the reconstructed scatterer. If scatterer $P$ in the point cloud has an amplitude value $q$ greater than the threshold value, it constitutes point cloud $C$, which is the filtered point cloud.

$$P \in C, \; q_P > q_{\text{threshold}} \tag{3}$$

The advantages of amplitude filtering are its simplicity, effectiveness, and low computation cost. However, determining the filtering threshold can be difficult. In addition, since the scattering coefficients of different regions or structures vary greatly, using the same filtering threshold will lead to uneven filtering results. The filtering effect is also poor in areas with strong scattering, while over-filtering will occur in areas with weak scattering. Scatterers with lower amplitude, for example, scatterers belonging to the ground, will be filtered out by mistake.

### 2.2.2. Confidence Filtering

Confidence filtering is designed to obtain the confidence value by calculating the correlation between the reconstruction result and the observation matrix in the process of compressed sensing and filter the whole point cloud according to this value. The confidence value is also directly related to the residual in the reconstruction process, with smaller residuals indicating better reconstruction results and higher confidence values. During the filtering process, these scatterers in the point cloud are easier to retain, while others are filtered out [21,22].

According to Equation (1), the expression of confidence $C_{i,j}$ is defined in the following way. $\varphi$ is the column vector in the observation matrix corresponding to the non-zero elements of the estimated vector $\sigma$. Note that the non-zero element of this vector means that there exists a scatterer at this height. $g_{i,j}$ is the sampling vector composed of multi-orbit complex images. According to this formula, the range of the confidence metric is between 0 and 1, whereby the large value indicates the high confidence of this point after reconstruction. That is to say, a scatterer is likely to be correctly obtained with larger probability.

$$C_{i,j} = \frac{\left| \varphi' \cdot g_{i,j} \right|}{\left| \varphi \right| \cdot \left| g_{i,j} \right|}, \varphi = \Phi_{\sigma_{i,j} \neq 0} \tag{4}$$

The overall effect of confidence filtering is better, especially in regions which have high confidence with relatively simple environment or terrain, so their overall shape is better preserved. However, the confidence level is low in regions with complex terrain. In complex regions, point clouds will thus be greatly affected by confidence filtering and most scatters will be over-filtered.

### 2.2.3. K Nearest Neighbor Filtering

K Nearest Neighbor Filtering (KNN) [23], also known as statistical filtering, is mainly used to remove outliers, which is characterized by sparse distribution in space. The algorithm calculates the average distance $r_k$ from each point to its nearest $K$ points. Then $r_k$ corresponding to all points in the point cloud should conform to a Gaussian distribution. The mean value $\overline{R}$ can be calculated according to $r_k$. The scatterer whose average distance $r_k$ is larger than the given variance value will be eliminated. This algorithm is effective in removing outliers with large density differences.

In addition, the advantage of KNN filtering is that it is not affected by the amplitude and the degree of confidence of the point cloud. For a point $p_i$ in point cloud $C$, the nearest $K$ point named $p_{ik}$ and the distance between two points is expressed as:

$$r_{ik} = \| p_i - p_{ik} \| \tag{5}$$

Then calculate the average distance between the $K$ points and $p_i$ to measure the distance between a point and its adjacent points.

$$d_i = \frac{1}{K} \sum_{k=1}^{K} r_{ik} \tag{6}$$

The points whose $d_i$ is greater than the threshold value is considered as noise points and need to be filtered out from point cloud $C$.

KNN filtering can retain the most complete terrain surface in several filtering methods and has good visual effect. However, for more complex terrain and noisy areas, the filtering ability is insufficient, and it is difficult to eliminate large continuous noisy scatterers.

### 2.2.4. Classification and Analysis of Point Clouds

Considering the properties of the point cloud obtained by TomoSAR, the scatterers before filtering can be roughly divided into the following four types. The full image of the selected point cloud is referred to in Section 3.

(1)    A scatterer in the correct position with large amplitude and confidence is the most ideal scatterer and should be retained during filtering.
(2)    A scatterer that is in the correct position but has smaller amplitude or confidence or both. This kind of point is usually located in the position with complex terrain, so it should be retained as much as possible after filtering.

By comparing the amplitude filtering results in Figure 2a and the KNN filtering results in Figure 2b, it is obvious that some points in the red circle are missing due to over-filtering.

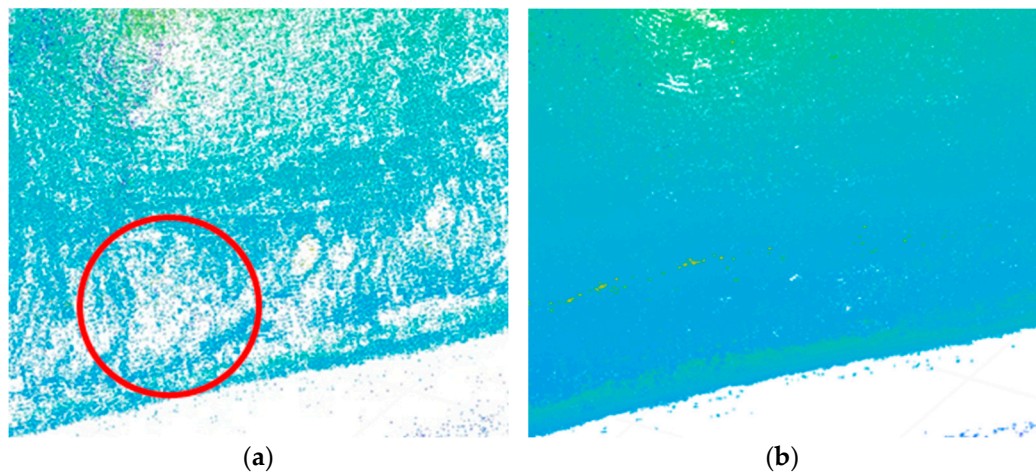

(a)                                                    (b)

**Figure 2.** (**a**) Part of Genhe amplitude filtering results. (**b**) Part of Genhe KNN filtering results.

(3)    Independent single noisy scatterer in the wrong position, which has larger amplitude or confidence. These noise points are randomly distributed in the point cloud.

Isolated noisy scatterers randomly distributed after confidence filtering are illustrated in Figure 3.

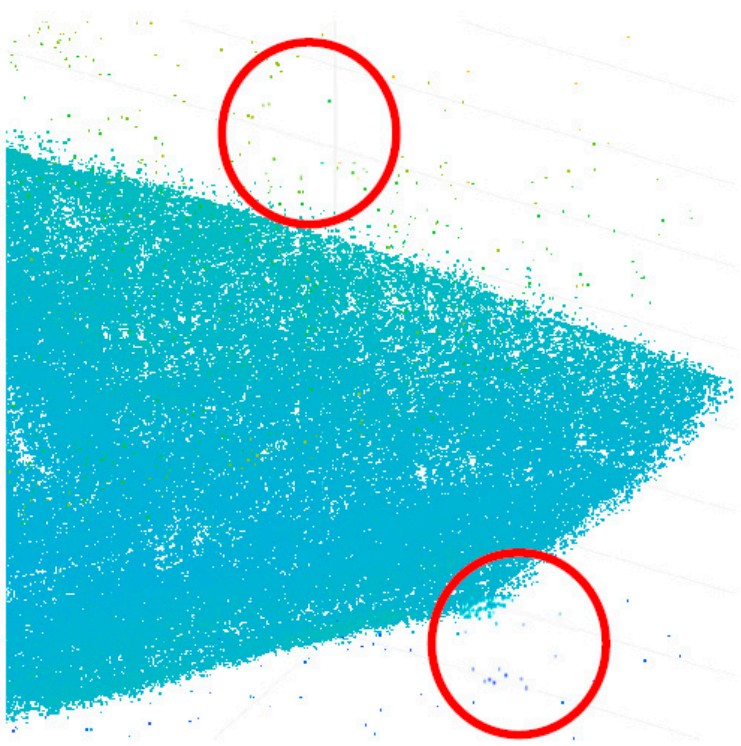

**Figure 3.** Part of Genhe confidence filtering results.

(4)     A large amount of continuous noise points in the wrong position with medium amplitude and low confidence. This kind of noise is usually caused by the elevation ambiguity of the imaging area, baseline error, low signal-to-noise ratio of the data, etc. They are usually distributed at the top and bottom or at a corner of the point cloud.

As is shown, the continuous noise in the red circle in Figure 4 is more obvious in the KNN filtering results.

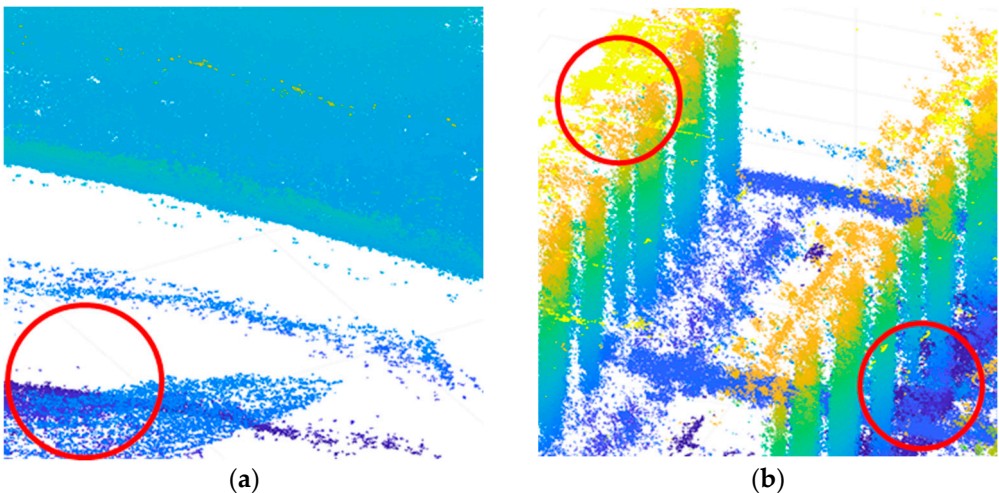

(**a**)                                                                                      (**b**)

**Figure 4.** (**a**) Part of Genhe KNN filtering results. (**b**) Part of Yuncheng KNN filtering results.

Based on the four types of scatterers present in the point cloud, the advantages and disadvantages of the three filtering methods can be systematically summarized. Amplitude filtering has the capability to retain all scatterers belonging to type one and most type two scatterers. It has a filtering effect on type four, but proves ineffective for type three. Similarly, confidence filtering is similar to amplitude filtering, with the ability to retain type

one and some type two scatterers in the point cloud. It exhibits good filtering effects for type four, but poor performance for type three. In contrast, K nearest neighbor filtering can retain type one and type two scatterers, and has a good filtering effect for type three, while the former two filtering algorithms fail to achieve results. However, it retains type four noisy scatterers as its disadvantage.

### 2.2.5. K Nearest Neighbor Comprehensive Weighted Filtering

Taking into consideration the strengths and limitations of the three aforementioned filtering algorithms, and in light of the practical considerations associated with 3D point cloud reconstruction, this paper proposes a novel filtering algorithm. Building upon the K nearest neighbor filtering approach, our proposed method assigns each point in the point cloud with a weight that reflects its amplitude and confidence. During the calculation of K nearest neighbor points, the relative distance is weighted, leading to the development of the K nearest neighbor comprehensive weighted filtering method. The distance formula for each point is expressed as follows:

$$d_i = \frac{1}{K} \sum_{k=1}^{K} [r_{ik} + (1 - G_k)Wg_k - A_k Wa_k] \tag{7}$$

In the above equation, $G_k$ and $A_k$ respectively represent the normalized values of confidence and amplitude, $Wg_k$ and $Wa_k$ respectively represent their corresponding weights, and $r_{ik}$ represents the Euclidean distance from point $i$ to its $k$-th nearest neighbor point. For each scatterer in the point cloud, its corresponding weighted distance $d_i$ will be calculated and it will be filtered out as noisy scatterers if the distance is greater than the threshold value. After this manipulation, the preserved scatterers constitute the filtered result.

According to the classification of scatterers mentioned above, for type one and type two, the point can be effectively retained since the scatterers are in the correct position, the amplitude and confidence of the K nearest neighboring points selected is large, and the weighted relative distance is smaller than the filtering threshold. For type three, although it has a larger amplitude and confidence, the scatterer is far away from its neighboring points and the weighted distance is larger than the threshold value, so it will be effectively filtered out. For noise of type four, the distance of the K nearest neighboring points selected is small, but the amplitude and confidence of these points is relatively low, and a large relative distance will be obtained after weighting, so they will also be filtered out. In summary, the proposed filtering algorithm has a better filtering effect over the aforementioned three methods.

### 2.3. Quantitative Evaluation of the Point Cloud

At present, the most commonly used algorithms for quantitative evaluation of point clouds are the Hausdorff distance (HD), the largest common point set (LCP), and the root mean square error (RMSE) [11,24]. Although these algorithms can realize an effective evaluation of the three-dimensional reconstruction results of TomoSAR, they have very serious limitations; that is, they require the model of the target as ground truth. In most practical application scenarios, the ground truth of the observation target is usually unknown and difficult to obtain.

Therefore, it is necessary to find a metric that does not rely on prior information to quantitatively evaluate the performance of various point cloud filtering algorithms. This metric needs to be quantifiable and independent. Currently, image entropy is used to evaluate the sharpness of the image. Small image entropy indicates high sharpness and vice versa. In previous work [6], the structural characteristics of point clouds were described by information theory, and the definition of image entropy has been extended to three-dimensional space. The denoising results can be evaluated quantitatively by calculating the three-dimensional entropy of point clouds after filtering. The smaller the

3D entropy, the higher the 3D reconstruction accuracy of the algorithm. The expression of the three-dimensional entropy of a point cloud is as follows:

$$H(x) = -\sum_{i=1}^{n} P(a_i) \cdot \log P(a_i) \tag{8}$$

$P(a_i)$ is the frequency of the occurrence of a single point cloud feature or feature group defined by the three-dimensional entropy algorithm. The commonly used features include point cloud amplitude, point cloud confidence, and the number of neighboring points within a certain distance of the point cloud.

The more the feature types that are selected when calculating 3D entropy, the lower the frequency of the same feature occurrence and the slower the operation speed. Therefore, when calculating 3D entropy, we should select appropriate feature types according to different needs and the total number of points in the point cloud. At present, considering that the visual effect of the point cloud only depends on the spatial structure between points, only the number of nearest points within a certain range is used as the criterion when calculating the three-dimensional entropy.

In order to calculate the number of nearest neighbor points of the target point, it is necessary to first determine a range centered on the target point, and the points within this range are the nearest neighbor points of the target point. In order to facilitate the coding operation, the cube range with the target point as the center and the radius of $R$ is adopted. The value of $R$ will also have a certain impact on the three-dimensional entropy results. Too small or too large $R$ values will tend to make the characteristics of points consistent, leading to a decline in the ability to quantitatively evaluate point clouds. When $R$ is difficult to determine, we draw multiple three-dimensional entropy curves with $R$ as the independent variable, which can evaluate the quality of the point clouds quantitatively.

Compared with other entropy-based algorithms, this algorithm has the advantage of good universality. It can be directly applied to the quantitative evaluation of various point clouds without grid, normalization, and other preprocessing operations. The calculated three-dimensional entropy value can additionally directly reflect the visual effect of the point cloud.

*2.4. Processing Framework*

In this section, we summarize the process from radar raw data to 3D reconstruction results, and our results are obtained through this framework. First, we perform two-dimensional BP imaging on the original data of each track. Then registration algorithms are applied in order to realize sub-pixel registration. According to the established 3D reconstruction geometric model, the compressed sensing algorithm is used to extract the scatterer's back-scattering coefficient and elevation information and further obtain the 3D reconstruction results. In the following, it is necessary to use 3D point cloud filtering algorithms to filter the original point cloud obtained from reconstruction and calculate the 3D entropy value of the filtered point cloud. Finally, the filtering performance of the aforementioned algorithms are compared and corresponding analyses are presented based on the 3D entropy. Here we adopt simulated data and two measured datasets to evaluate the performance. The overall flowchart is shown in Figure 5.

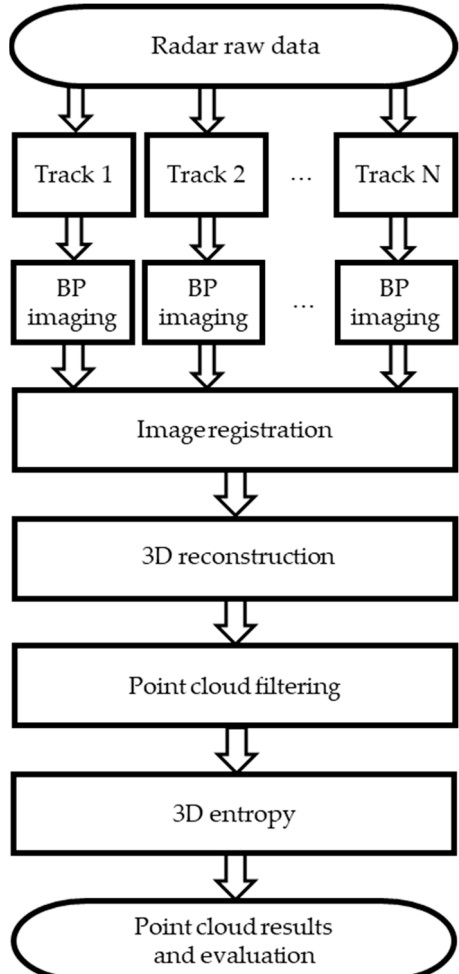

**Figure 5.** Process flowchart for 3D point cloud filtering.

## 3. Results

### 3.1. Simulated Data

The building in this simulation is 20 m high, 10 m wide, and 50 m long. The simulation configuration is listed in Table 1. The simulation model of the building is shown in Figure 6. The 2D imaging result of the simulated building is shown in Figure 7.

**Table 1.** Simulation configuration of imaging.

| Parameter | Value |
| --- | --- |
| Carrier Frequency | 1.5 GHz |
| Bandwidth | 200 MHz |
| Slant Range | 3000 m |
| Number of Tracks | 11 |
| Track Spacing | 10 m |
| Angle of View | 45° |

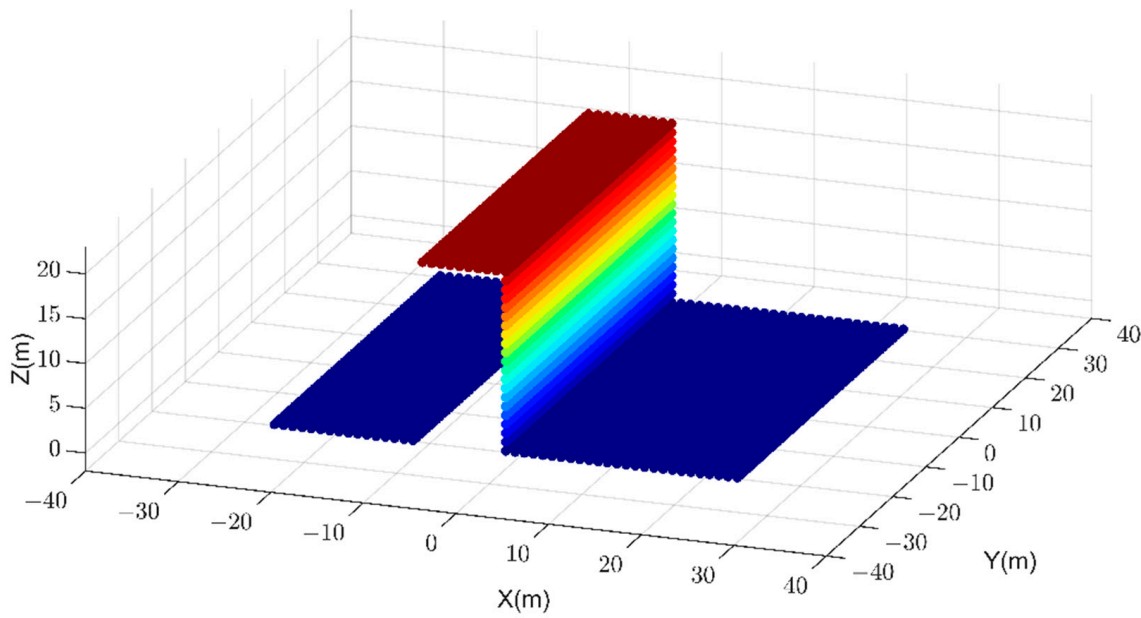

**Figure 6.** Simulation model of the building.

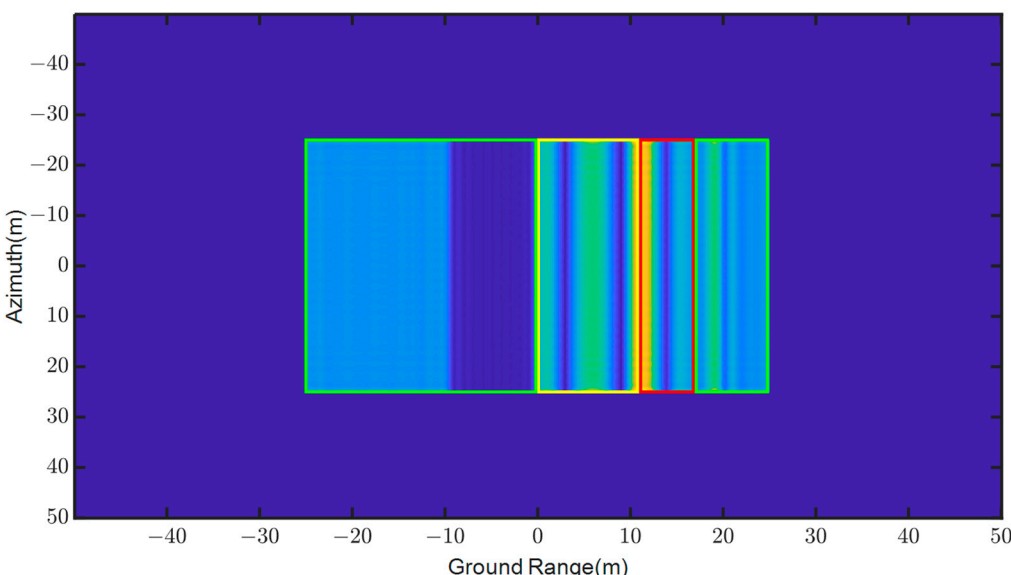

**Figure 7.** 2D imaging result of the simulated building.

In the 2D imaging result of the simulated building, the green box represents the ground area, the yellow box contains the ground area and building facade, and the red box contains the ground, building facade, and roof.

In the reconstruction of simulated data, amplitude error and phase error are randomly added to the data obtained from different orbits to simulate the amplitude and phase error in the radar sampling process. In addition, Gaussian noise with a signal-to-noise ratio of 10 dB is added to each two-dimensional complex image. The three-dimensional reconstruction of the simulated building is then realized using the compressed sensing algorithm. The original point cloud after reconstruction is shown in Figure 8, where the color represents the elevation of the three-dimensional reconstruction result.

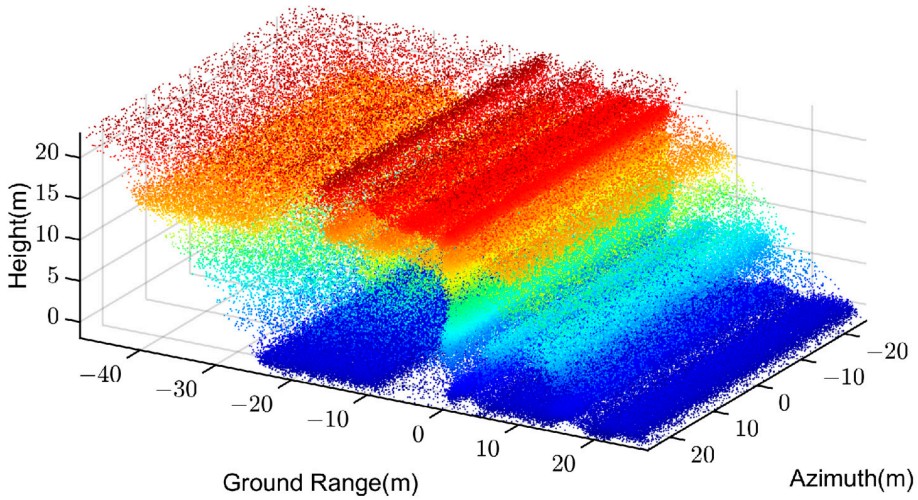

**Figure 8.** Original 3D point cloud of the building simulation.

Then four different filtering methods mentioned are adopted to filter the original point cloud, and the results are illustrated in Figures 9–12.

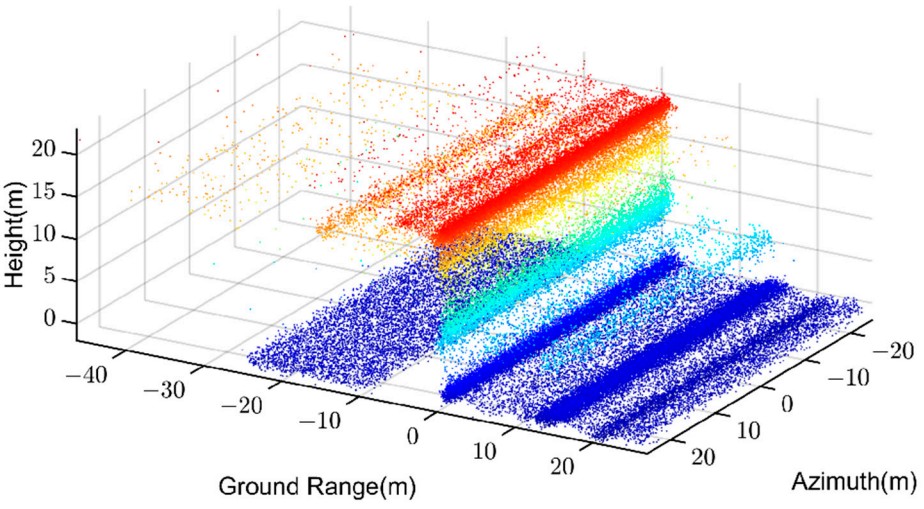

**Figure 9.** Simulated 3D point cloud filtered by Amplitude.

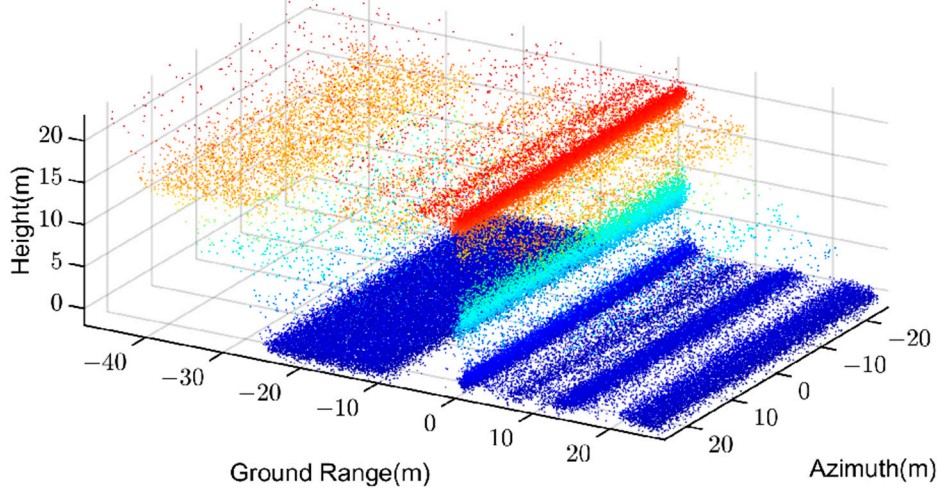

**Figure 10.** Simulated 3D point cloud filtered by Confidence.

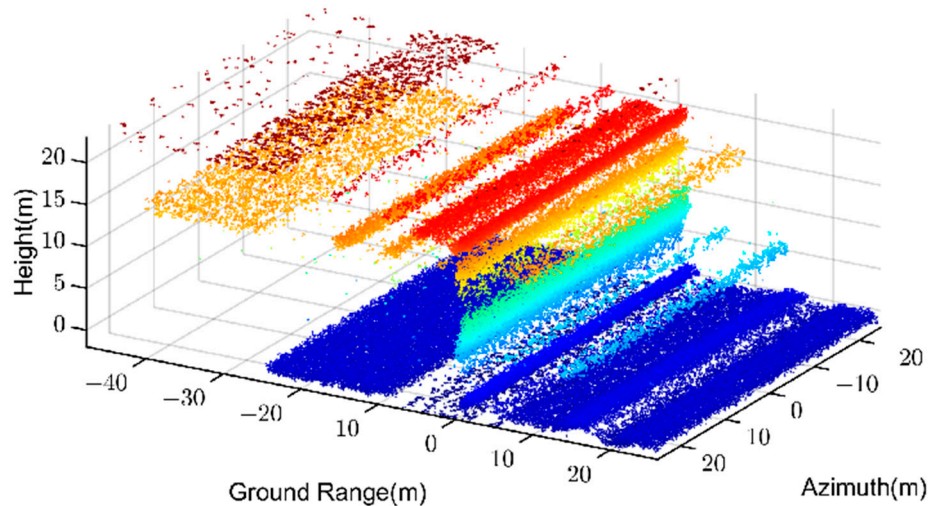

**Figure 11.** Simulated 3D point cloud filtered by KNN.

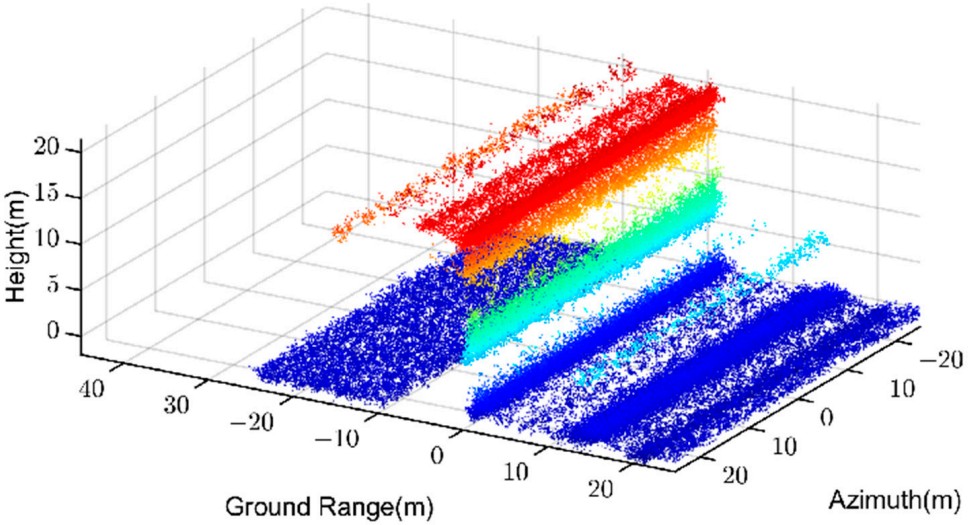

**Figure 12.** Simulated 3D point cloud filtered by KNN-weight.

The reconstructed point cloud in Figures 8–10 shows that there are serious layover problems. The method we proposed aims to realize tomogram filtering with the aid of the scatterers' attributes and the spatial structure information of the point cloud. The proposed method itself does not have the ability to improve the scatterer unmixing performance. However, due to the fact that the erroneous scatterers generated by layover or blurring have certain characteristics in the point cloud, a piece of scatterers for example distributes at the top and bottom of the point cloud with medium amplitude and low confidence (point cloud type four mentioned in Section 2.2.4). The proposed method can filter out such scatterers to a certain extent and preserve the main structure of the building, achieving the goal of improving the quality of the point cloud.

Among the four filtering results in Figures 9–12, the K nearest neighbor comprehensive weighted filtering algorithm has a better filtering effect on the continuous noise points on the left and on the isolated noise points around the building. In addition, it preserves the structure of the building as much as possible. As shown in Figure 13, the superiority of this filtering algorithm can also be proved by the 3D entropy result of the point cloud.

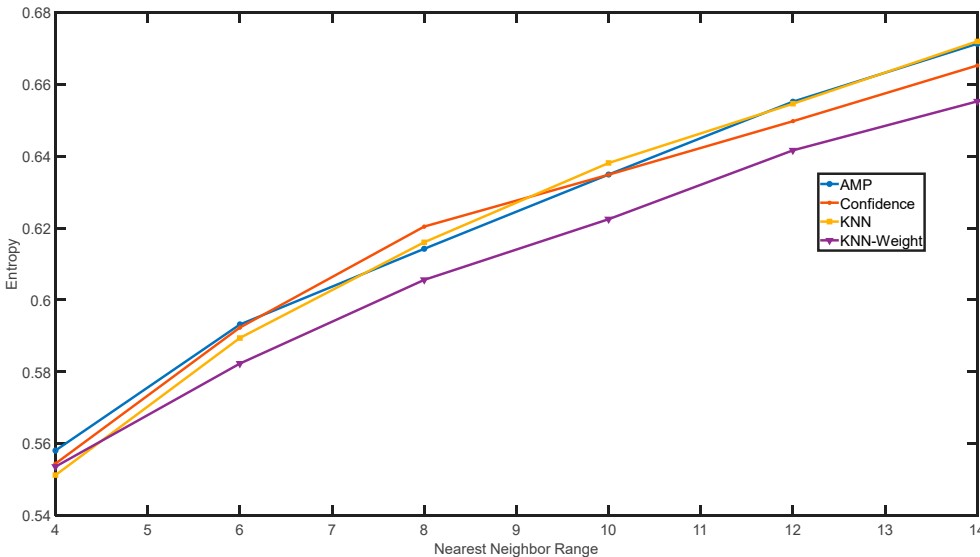

**Figure 13.** 3D entropy results of the simulated 3D point cloud.

*3.2. TomoSAR Data Processing Results*

In order to further validate the proposed method, we collected two measured datasets, i.e., a mountainous area of Genhe, Nei Monggol Autonomous Region and an urban area of Yuncheng, Shanxi province. Tomograms corresponding to these two areas are filtered with the four filtering methods mentioned in the previous section for comparison and analysis. The optical and SAR images of both regions are shown in Figures 14 and 15.

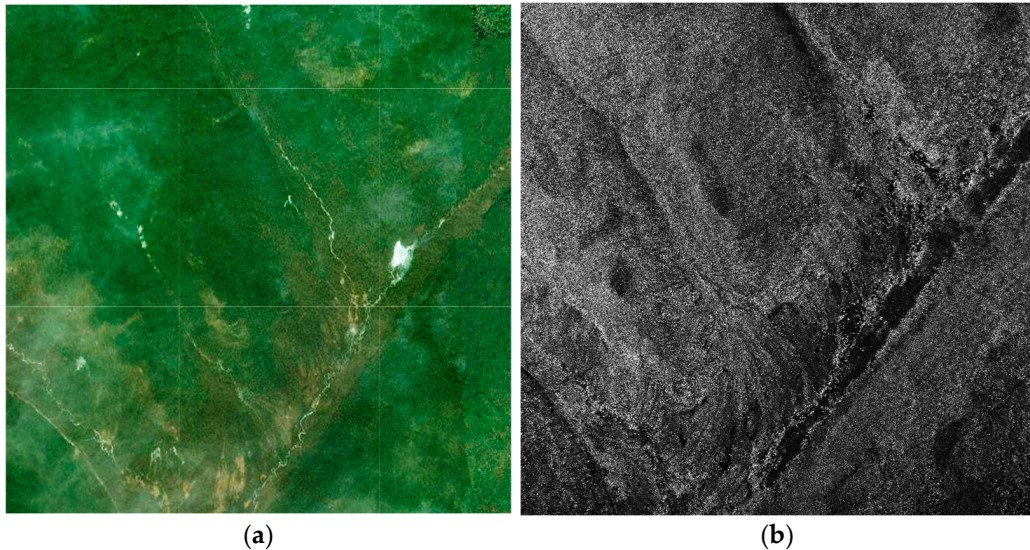

(**a**)                                         (**b**)

**Figure 14.** (**a**) Genhe optical image. (**b**) Genhe SAR image.

3.2.1. Original Point Cloud Data after 3D Reconstruction

The original point cloud has a large number of noisy scatterers. Among these noises, the isolated noise points generated by random noise are widely distributed in the two tomograms. Additionally, regarding Genhe, the continuous noise generated by image blurring is mainly distributed in the part below the ground, presenting a horizontal distribution. As for the tomogram of Yuncheng, the noisy scatterers are distributed at the bottom of the building structure, showing an inclined distribution. The original 3D point cloud of Genhe and Yuncheng are shown in Figures 16 and 17.

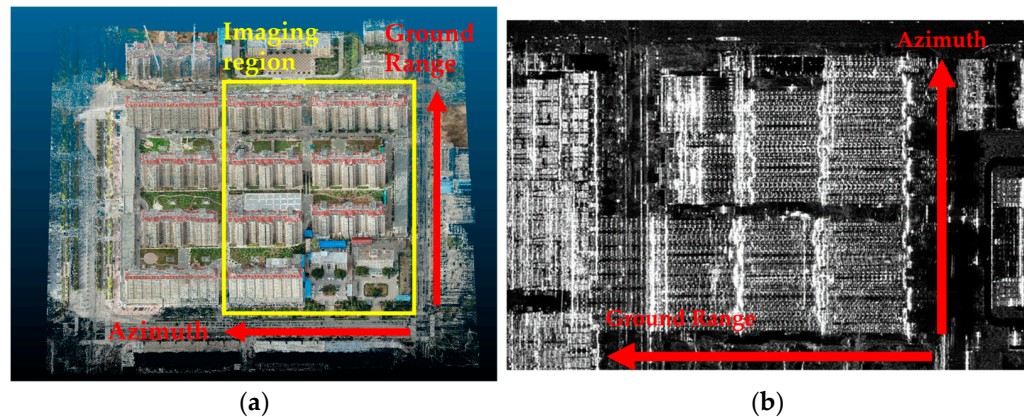

**Figure 15.** (**a**) Yuncheng optical image. (**b**) Yuncheng SAR image.

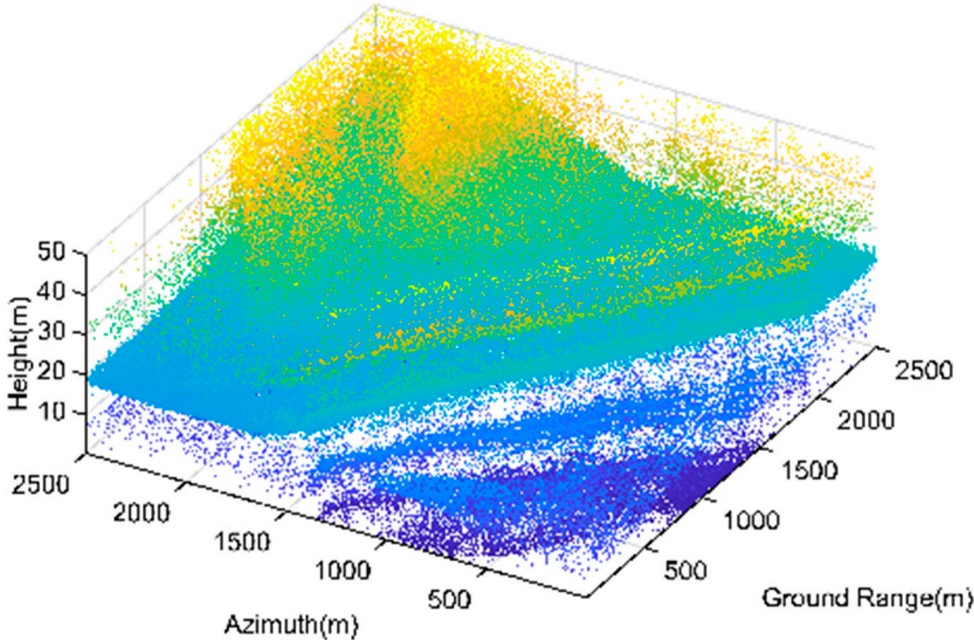

**Figure 16.** Genhe Original 3D Point Cloud.

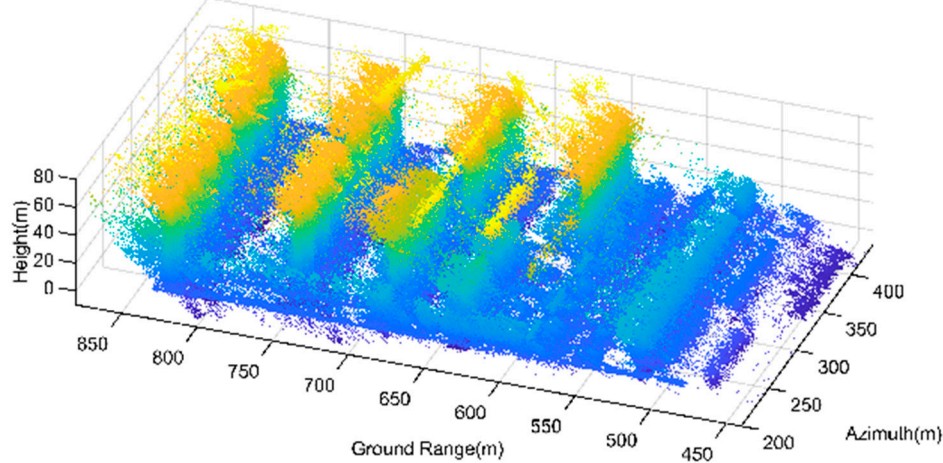

**Figure 17.** Yuncheng Original 3D Point Cloud.

### 3.2.2. Amplitude Filtering

Amplitude filtering has good universality and can filter out most of the noise. It can preserve the overall shape of the terrain and the buildings. However, the continuous noise filtering effect under the terrain and at the bottom of the building is not optimal, and some continuous terrain and building facades may be over-filtered. In the filtering results of the Genhe point cloud, the ground that should have been continuous has been spoiled by some holes due to over-filtering, and the continuous noise under the terrain has not been effectively filtered. The 3D point cloud filtered by amplitude of Genhe and Yuncheng are shown in Figures 18 and 19.

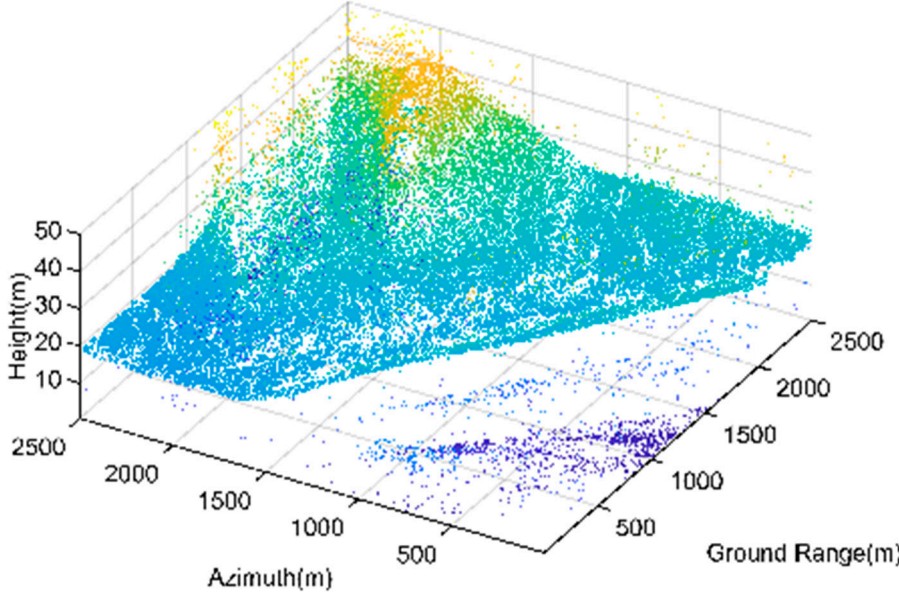

**Figure 18.** Genhe 3D point cloud filtered by Amplitude.

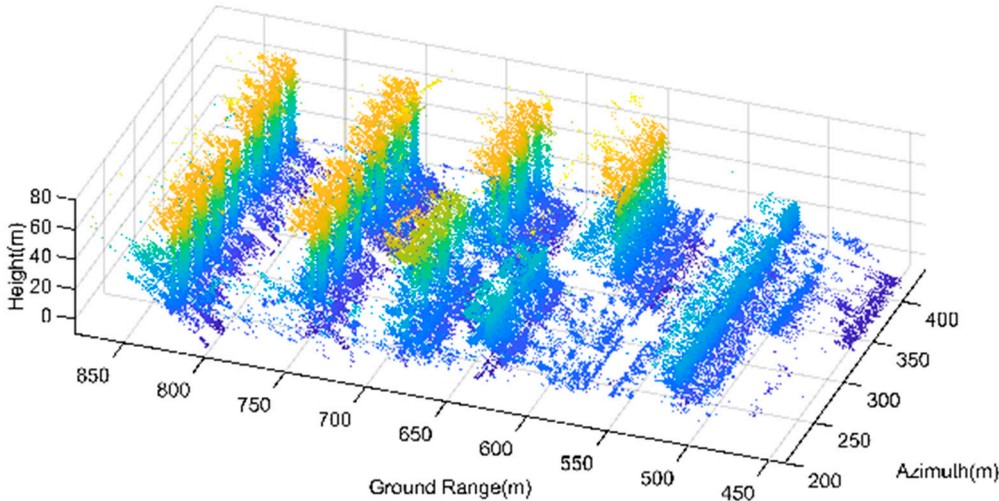

**Figure 19.** Yuncheng 3D point cloud filtered by Amplitude.

### 3.2.3. Confidence Filtering

Confidence filtering aims to address the limitation of amplitude filtering. Filtering the point cloud from the perspective of the 3D reconstruction principle can effectively eliminate the continuous noise caused by blurring. It can be seen from the filtered tomogram that this part of the noise is effectively eliminated, and the terrain and overall structure of the building are complete, leading to a better visual effect. However, confidence filtering still cannot eliminate the isolated noise points in the point cloud, and the overall confidence

in the area with complex terrain (such as the left side of the Genhe point cloud) is low, which will also lead to the phenomenon of over-filtering. The 3D point cloud filtered by confidence of Genhe and Yuncheng are shown in Figures 20 and 21.

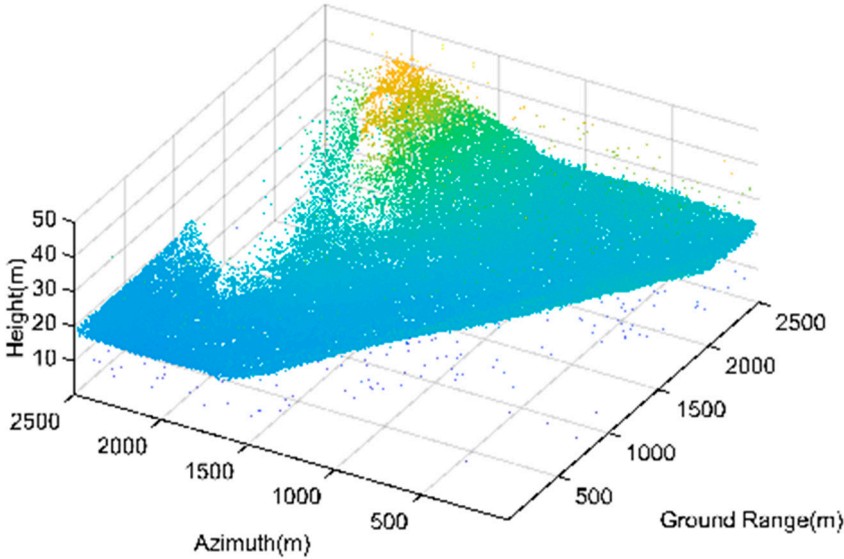

**Figure 20.** Genhe 3D point cloud filtered by Confidence.

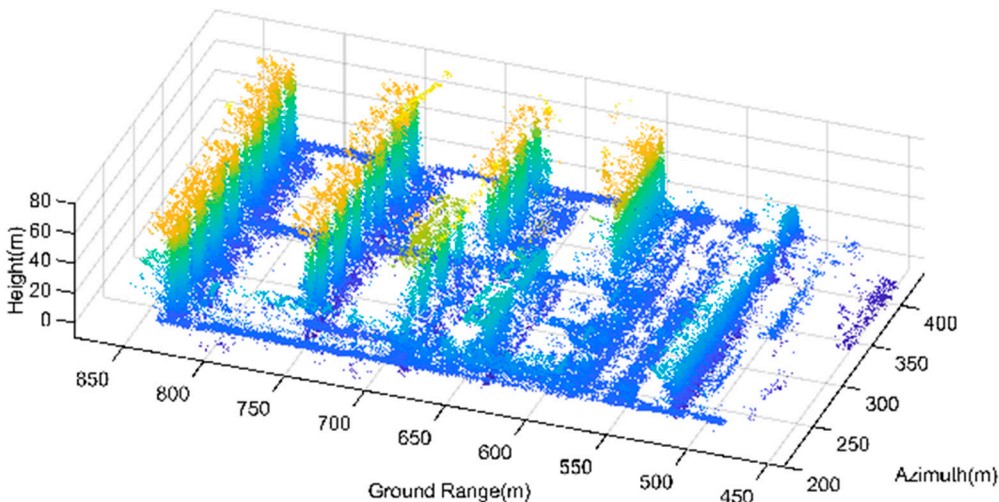

**Figure 21.** Yuncheng 3D point cloud filtered by Confidence.

### 3.2.4. KNN Filtering

KNN filtering mainly aimed at eliminating the isolated noisy scatterers randomly distributed in the point cloud, while preserving the complete target structure. In the KNN filtering results, most of the isolated noise points have been eliminated, but a large amount of continuous noise still exists. The scatterers at the bottom of the tomogram of Genhe and the scatterers in the top and bottom of the tomogram of Yuncheng are typical continuous noisy scatterers. This is also a limitation of KNN filtering. The 3D point cloud filtered by KNN of Genhe and Yuncheng are shown in Figures 22 and 23.

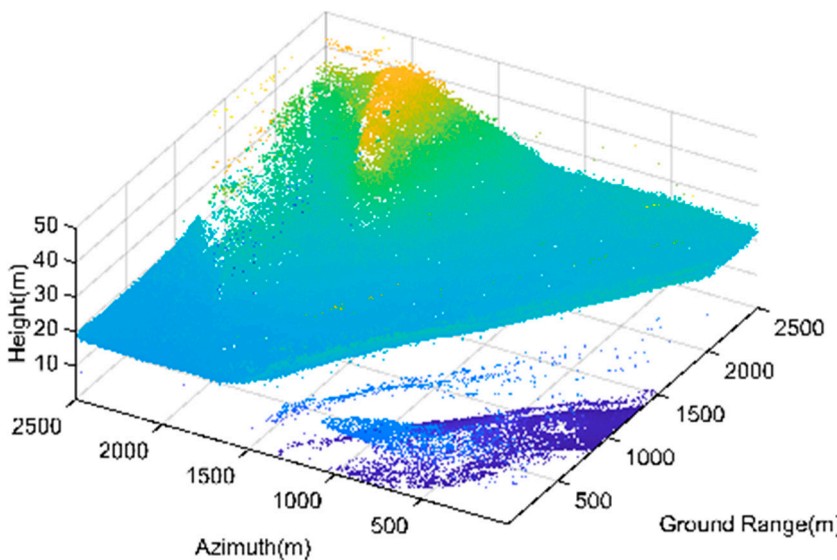

**Figure 22.** Genhe 3D point cloud filtered by KNN.

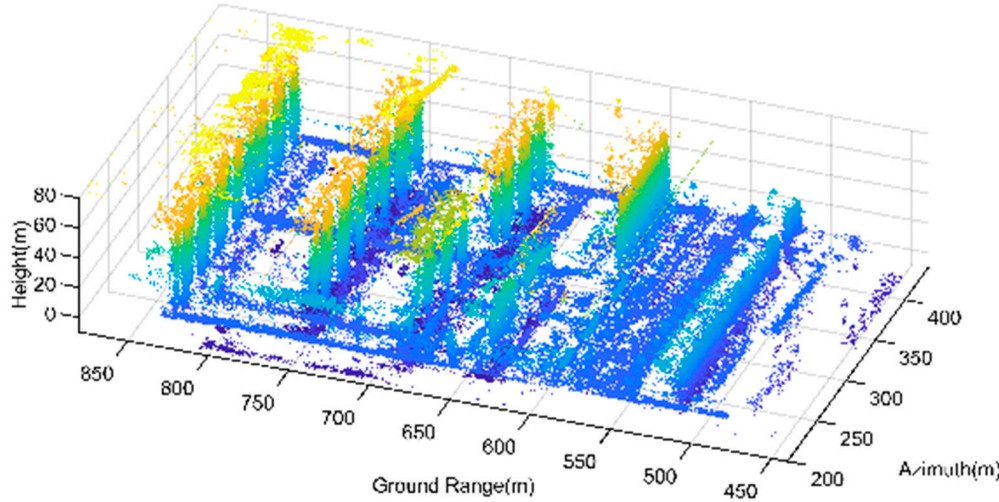

**Figure 23.** Yuncheng 3D point cloud filtered by KNN.

### 3.2.5. K Nearest Neighbor Comprehensive Weighted Filtering

The K nearest neighbor comprehensive weighted filtering algorithm not only retains the advantages of K nearest neighbor filtering itself, such as preserving complete terrain surfaces and effectively filtering independent noise points, but also filters the noise points in these areas by taking advantage of the low amplitude and the confidence of continuous noise points. Generally, this algorithm has a good point cloud filtering effect and represents a promising approach for improving point cloud quality. The 3D point cloud filtered by K nearest neighbor comprehensive weighted filtering method of Genhe and Yuncheng are shown in Figures 24 and 25.

For the purpose of better comparison, a region is selected from the processing result of Yuncheng. Four filtering methods are respectively applied to process this region with each resulting point cloud controlled to have 100 K points. The results are shown in Figure 26.

### 3.2.6. Three-Dimensional Entropy Result

The three-dimensional entropy values calculated by each point cloud filtering algorithm under different neighboring ranges are compared, and the results are shown in Figure 27:

With the increase in the value of the nearest neighbor range, the entropy value increases gradually. When the value of the nearest neighbor range is small, the number of nearest neighbor points is concentrated, and the results of several filtering algorithms have similar entropy values. When the value of the nearest neighbor range increases, the number of the nearest neighbor points of the filtering results with more noise around will have a uniform distribution, which will increase the entropy value. On the contrary, point clouds having less noise and a more concentrated distribution will have a smaller entropy. Therefore, the point cloud filtering results can be quantitatively analyzed by comparing the three-dimensional entropy in different neighborhood ranges.

A line chart to describe the relation between the neighborhood range and the 3D entropy value is drawn by calculating the 3D entropy values of the four kinds of filtered point clouds with different neighborhood ranges. The figure shows that, in most cases, especially when the value is large, the three-dimensional entropy value of the point cloud obtained by the K nearest neighbor comprehensive weighted filtering algorithm is small. This indicates that the point cloud is distributed more intensively, which can better reflect the shape of the target object.

It is worth mentioning that in Figure 28, the confidence-based filtering method outperforms other methods. The reasons are twofold. The first is that the data quality of the Genhe region is good, i.e., the SNR of the image is high. In fact, the confidence value is related to the signal-to-noise ratio of the image. Therefore, the inherent characteristics of this dataset determine that the confidence filtering outperforms other methods. The second is that the region in Genhe is a wide mountainous area with less layover, plus the sparsity is set to one for three-dimensional reconstruction, which will decrease the number of noisy scatterers. Therefore, the confidence filtering in this area has a good filtering effect, reflected in the lower three-dimensional entropy value compared to other methods. The 3D entropy results of range fourteen are listed in Table 2.

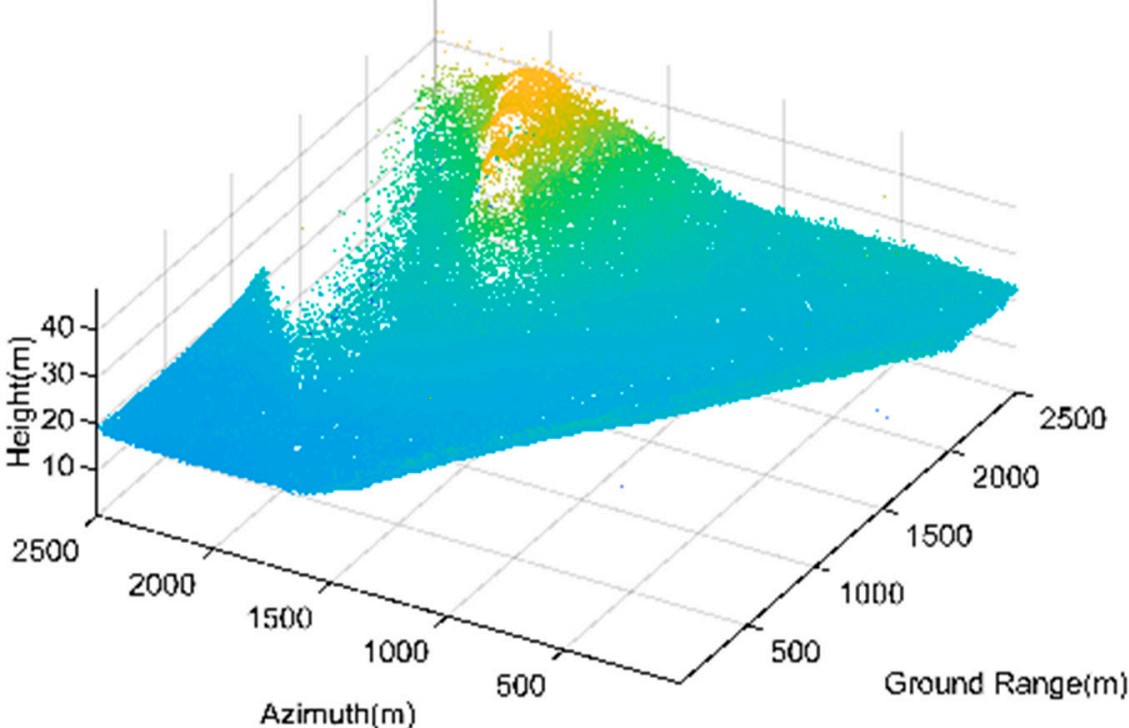

**Figure 24.** Genhe 3D point cloud filtered by KNN-weight.

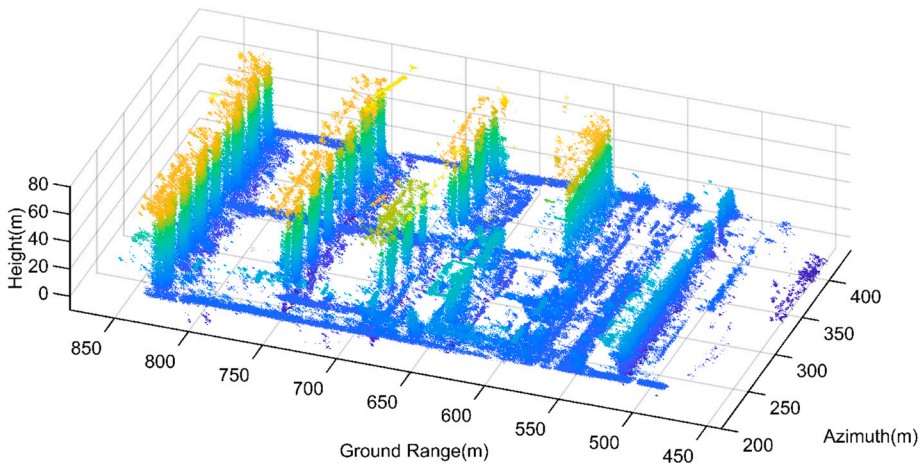

**Figure 25.** Yuncheng 3D point cloud filtered by KNN-weight.

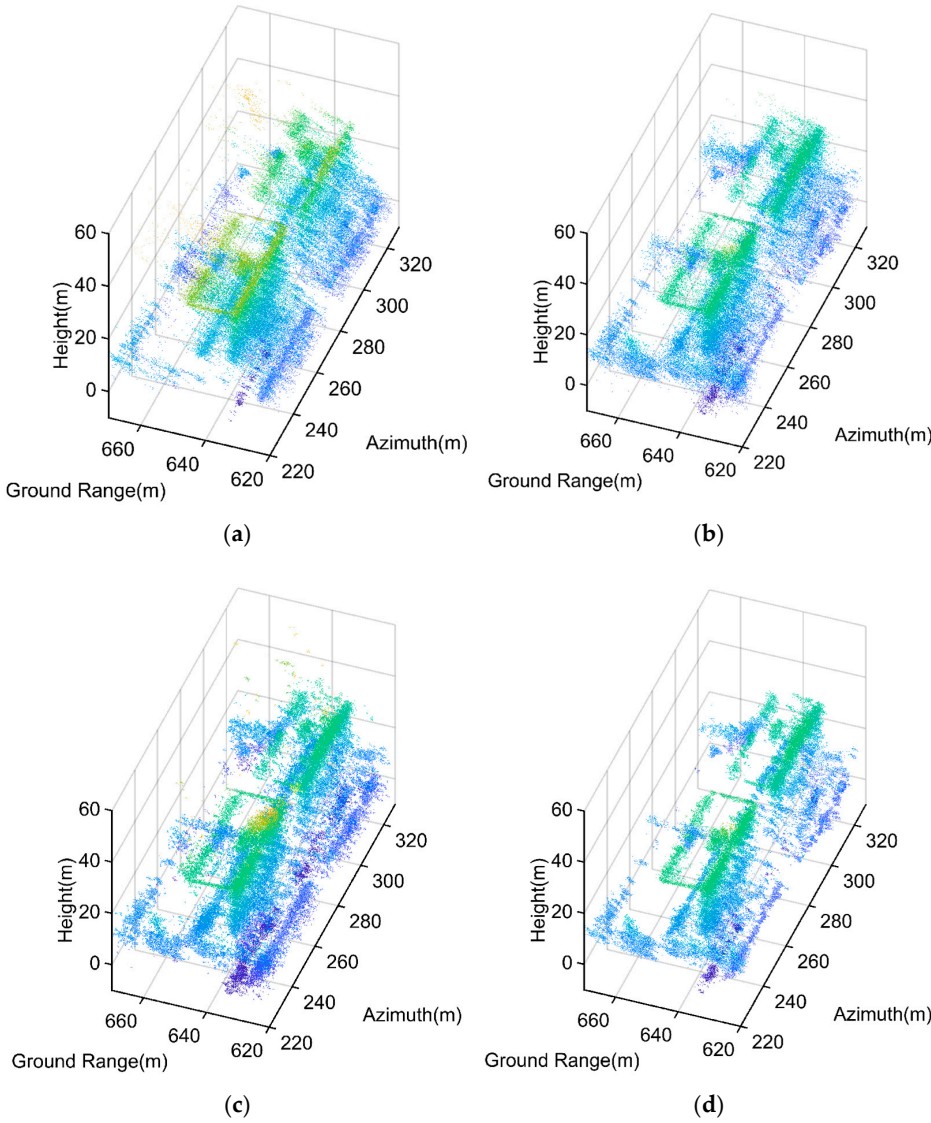

**Figure 26.** (**a**) Yuncheng point cloud detail filtered by Amplitude. (**b**) Yuncheng point cloud detail filtered by Confidence. (**c**) Yuncheng point cloud detail filtered by KNN. (**d**) Yuncheng point cloud detail filtered by KNN-weight.

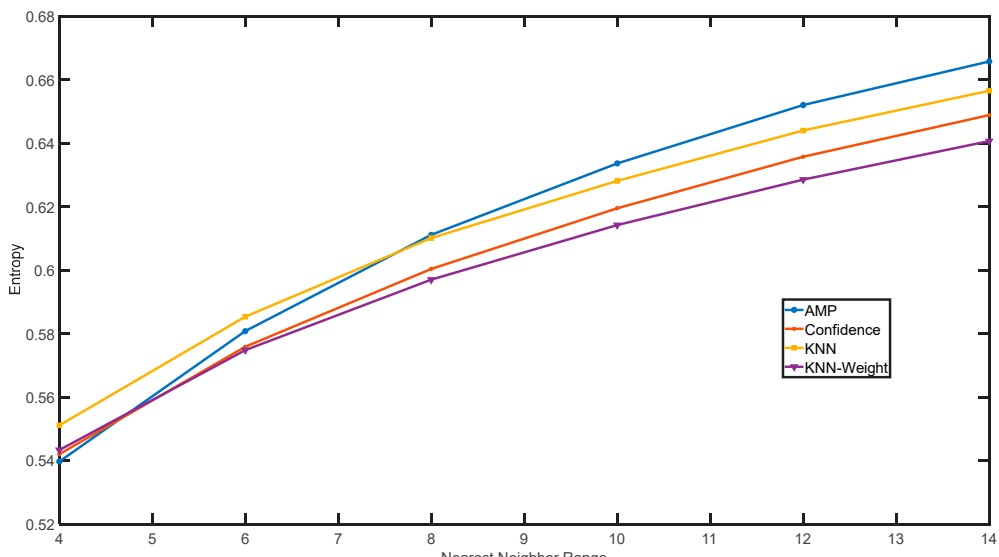

**Figure 27.** 3D entropy results of the Yuncheng point cloud.

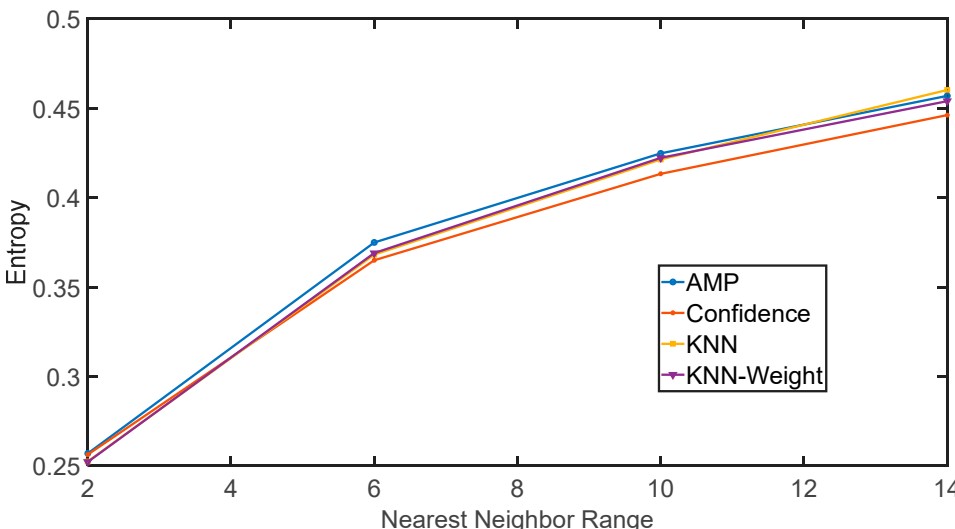

**Figure 28.** 3D entropy results of the Genhe point cloud.

**Table 2.** 3D entropy results of range fourteen.

| Filtering Method | Simulation Building | Yuncheng | Genhe |
|---|---|---|---|
| Amplitude | 0.6713 | 0.6658 | 0.4569 |
| Confidence | 0.6652 | 0.6489 | 0.4462 |
| KNN | 0.6719 | 0.6566 | 0.4603 |
| KNN weight | 0.6553 | 0.6407 | 0.4540 |

## 4. Discussion

Based on the above experimental results and analysis, we summarize the advantages of the proposed filtering method of the 3D point cloud. Its potential shortcomings and solutions are also given in the following way.

The advantages of the proposed method can be summarized as universality which includes two aspects: the scene types and the noise types.

In terms of the universality of the scene types, the point cloud filtering method we introduced does not require prior information of the observed scene, but instead utilizes the parameters of the scatterers in the point cloud and the spatial structure information

between them. Therefore, it has adaptability and a good filtering effect on the point clouds obtained from the 3D reconstruction of SAR in various scenarios.

In terms of the universality of noise types, this method combines the advantages of various traditional filtering algorithms and has good filtering performance for common noise types in point clouds, such as continuous noise generated by radar amplitude and phase errors or blurring during the 3D reconstruction process and randomly distributed thermal noise. At the same time, it tries to preserve non-noise points as much as possible to avoid the occurrence of over-filtering.

We also discovered the shortcomings of the proposed method during data processing. Due to the use of more point cloud information in the proposed method, its processing efficiency may be lower than the traditional filtering methods. For small scale scenes and areas with relatively single noise types, there is not much difference between the filtering results obtained by selecting the corresponding filtering method or using the filtering method we proposed. However, large-scale scenes usually have two or more types of noise, and traditional filtering methods are not competent for various noise points simultaneously. Therefore, the filtering method we proposed shows more significant advantages over other methods, especially in large-scale scenes. In addition, although it makes some progress compared with the confidence filtering algorithm, with this method it is still difficult to avoid the problem of over-filtering in some complex areas.

The solutions for the above shortcomings may be achieved in the following way. We can try to combine the proposed filtering method with the 3D entropy algorithm and dynamically adjust the filtering algorithm based on the 3D entropy results. According to the definition of the comprehensive weighting algorithm, it can degenerate into several traditional filtering algorithms under different parameter values, thereby further improving its adaptability to different point cloud situations.

## 5. Conclusions

Synthetic aperture radar tomography solves the problem of layover in 2D SAR images in complex terrain. The 3D point cloud obtained by TomoSAR allows for better identification and processing of observed scenes. However, due to factors such as equipment, target structure, and noise, the reconstructed point cloud often suffers from numerous noisy scatterers. Therefore, it is necessary to use point cloud filtering to improve the quality of the 3D reconstruction.

This paper compares and analyzes the characteristics of three commonly used filtering algorithms for 3D point clouds in TomoSAR reconstruction. Based on the advantages of these methods, we propose a K nearest neighbor comprehensive weighted filtering algorithm. By comparing the filtering results of these four algorithms using simulated data and real TomoSAR datasets from Genhe and Yuncheng, the strengths and weaknesses of each algorithm are qualitatively analyzed. The 3D entropy of a point cloud, which is an extension of the image entropy to three-dimensional space, is obtained by combining the spatial structure of the point cloud. It is used to evaluate the filtering effect of the point cloud. These results validate that the proposed method outperforms existing point cloud filtering algorithms and prove its practicability.

**Author Contributions:** Conceptualization, Z.J.; Methodology, S.D.; Software, S.D.; Validation, Z.J. and L.Z.; Formal analysis, S.D. and Q.Y. (Qiancheng Yan); Investigation, S.D. and Q.Y. (Qiancheng Yan); Resources, Z.J. and Q.Y. (Qiancheng Yan); Writing—original draft, S.D.; Writing—review & editing, Z.J., L.Z. and Q.Y. (Qianning Yuan); Visualization, S.D.; Supervision, Z.J. and L.Z.; Project administration, L.Z. All authors have read and agreed to the published version of the manuscript.

**Funding:** This research was funded by the National Natural Science Foundation of China (No. 62101535, No. 61991420, No. 61991421, and No.41971329).

**Data Availability Statement:** The data presented in this study are available on request from the corresponding author.

**Conflicts of Interest:** The authors declare no conflict of interest.

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
