# Peer review of "A Novel Filtering Method of 3D Reconstruction Point Cloud from Tomographic SAR"

_remotesensing, doi:10.3390/rs15123076_

Round 1
Reviewer 1 Report
To solve the problem of large noise in 3D SAR point cloud reconstruction, this paper proposes a K-nearest neighbor comprehensive weighted filtering algorithm based on the strengths and limitations of amplitude filtering, confidence filtering and K-nearest neighbor filtering, and the filtered point cloud is evaluated quantitatively using three-dimensional entropy.
Overall, the article is innovative, with the following issues:
(1) In line 142, the subject is missing before “represents”.
(2) The reference to mk in formula (7) is unclear.
(3) The underline "itself" in line 421 should be dropped.
(4) In line 452, "Table 1" should be "Table 2"; it is suggested to add the three-dimensional entropy results of the Genhe region.
(5) It is suggested to change the background color of the 3D point cloud result picture to white.
(6) Whether the method is universal. For small data sets, the effect is not outstanding, and complex areas have the problem of over-filtering. It is suggested to state the application conditions of the method.
NO
Reviewer 2 Report
In this manuscript, the author introduced K-nearest neighbor comprehensive weighted filtering algorithm and three-dimensional entropy quantitative evaluation, trying to improve the quality of 3D reconstruction in SAR tomography. Based on the combination of three filtering methods and a predefined distance formula, the author shown good filtering effect for 4 types scatterers and obtained high-quality 3D point clouds.
But there are some flaws to be modified.
1. Was the 3D reconstruction based on PS points in this manuscript? In that case, the authors should explain the difference between point cloud filtering and PS point selection. For instance, the amplitude filtering can also be used before 3D reconstruction to select PS point, reducing computation cost for subsequent processing.
2. The idea of TomoSAR is to solve the problem of layover in 2D SAR images in complex terrain and the authors also mentioned this problem according to scatterers of type2. It is supposed to discuss that in results and discussion section. Besides, there may be serious layover problems in the simulation model in Figure 6.
3. The second chapter Methods is not complete, there is no description of the entire processing framework mentioned in line 106.
4. The author should give the SAR image of two study area before or in Figure 3, Figure 4, for clarify. The same as Figure 6.
5. Line 469-474 should be in chapter 3 rather than chapter 4.
6. In line 91 of page 2, “having…” should be “have…”.
In line 146 of page 5, “Tomo SAR…” should be “TomoSAR…”.
In line 159 of page 5, the expression of “In 2007…” is confusing.
Round 2
Reviewer 2 Report
All my questions are answered,but there still be a few minor issues to be revised in this manuscript.
1. For comment 2 in last review, the authors point out that the method proposed method itself does not have the ability to improve the scatterer unmixing performance and just provides a comprehensive filtering method from a new perspective. However, solving the problem of layover should be included in the processing of 3D-reconstruction rather than point cloud filtering.
Besides, the authors should give a more detailed explanation of “certain characteristics” in Page 14, Line370, it is confused.
2. The 2D imaging result of simulated building is lack of annotation and unable to match the subsequent processing results for its less elaborate imaging.
3. Figure 15 is lack of annotation of range and azimuth direction, meanwhile, it seems that the 3D point cloud result in Figure 17 is not correspond to Figure 15 for the number of buildings.
4. The author should reconfirm the size of Figure 8-13, they are too large.
5. The English level should be improved for a more clear presentation of this manuscript.
